# Pan-cancer circulating tumor DNA detection in over 10,000 Chinese patients

Yongliang Zhang[1,8], Yu Yao[2,8], Yaping Xu[3,8], Lifeng Li[3], Yan Gong[1], Kai Zhang[4], Meng Zhang[5], Yanfang Guan[3], Lianpeng Chang[3], Xuefeng Xia[3], Lin Li[6,7], Shuqin Jia[5✉] & Qiang Zeng [1✉]

Circulating tumor DNA (ctDNA) provides a noninvasive approach to elucidate a patient's genomic landscape and actionable information. Here, we design a ctDNA-based study of over 10,000 pan-cancer Chinese patients. Using parallel sequencing between plasma and white blood cells, 14% of plasma cell-free DNA samples contain clonal hematopoiesis (CH) variants, for which detectability increases with age. After eliminating CH variants, ctDNA is detected in 73.5% of plasma samples, with small cell lung cancer (91.1%) and prostate cancer (87.9%) showing the highest detectability. The landscape of putative driver genes revealed by ctDNA profiling is similar to that in a tissue-based database ($R^2 = 0.87$, $p < 0.001$) but also shows some discrepancies, such as higher *EGFR* (44.8% versus 25.2%) and lower *KRAS* (6.8% versus 27.2%) frequencies in non-small cell lung cancer, and a higher *TP53* frequency in hepatocellular carcinoma (53.1% versus 28.6%). Up to 41.2% of plasma samples harbor drug-sensitive alterations. These findings may be helpful for identifying therapeutic targets and combined treatment strategies.

[1] Health Management Institute, The Second Medical Center & National Clinical Research Center for Geriatric Diseases, Chinese PLA General Hospital, Beijing 100089, P. R. China. [2] Department of Medical Oncology, The First Affiliated Hospital of Xi'an Jiaotong University, Xi'an 710061 Shanxi, P. R. China. [3] Geneplus-Beijing Institute, Beijing 102206, P. R. China. [4] Department of Cancer Prevention, National Cancer Center/Cancer Hospital, Chinese Academy of Medical Sciences and Peking Union Medical College, Beijing 100010, P. R. China. [5] Department of Molecular Diagnostics, Key Laboratory of Carcinogenesis and Translational Research (Ministry of Education/Beijing), Peking University Cancer Hospital & Institute, Beijing 100142, P. R. China. [6] Department of Medical Oncology, Beijing Hospital, National Center of Gerontology, Beijing 100010, P. R. China. [7] Institute of Geriatric Medicine, Chinese Academy of Medical Sciences, Beijing 100010, P. R. China. [8]These authors contributed equally: Yongliang Zhang, Yu Yao, Yaping Xu. ✉email: shuqin_jia@hsc.pku.edu.cn; zq301@126.com

Genomic analysis has markedly changed the clinical management of several malignancies[1]. Several international consortia, including The Cancer Genome Atlas (TCGA) and MSK-IMPACT Clinical Sequencing Cohort (MSKCC)[2], have performed comprehensive genomic analysis of multiple tumor types. Such efforts dramatically enhance our understanding of the cancer genome and provide open access resources for cancer research worldwide[3]. However, these consortia primarily focus on profiling Caucasian patients and lack data from Asian patients, who account for ~50% and 70% of overall cancer incidence and mortality, respectively, according to Global Cancer Statistics in 2018[4]. Despite their being the largest population worldwide, Chinese patients still lack a comprehensive genomic resource. This bias in genomic research could result in inequitable tumor prevention and treatment strategies around the world. Thus clarifying the pan-cancer genomic landscape in Chinese patients could help to improve therapeutic regimens and survival[5].

Recently, noninvasive blood-based liquid biopsy and circulating cell-free DNA (cfDNA) analysis has provided another route to the cancer genome that overcomes issues with intratumor heterogeneity[6]. Although it took decades to clarify that tumor-specific genomic markers could be traced in cfDNA[7,8], this technique has seen rapid development in the past decade[9]. Tumor-derived cfDNA is detectable across multiple cancer types[10]. However, the broad insights gleaned from genomic information derived from cfDNA introduce additional complexities compared with tissue-based genomic profiling. The concordance and discrepancies between circulating tumor DNA (ctDNA) and tissue sequencing have been evaluated to some extent but may be confounded by temporal and spatial heterogeneity[11–14]. In addition, the detectability of ctDNA is limited by tumor size and the technological platform[15]. Taking these factors into consideration, ctDNA and tissue sequencing could be mutually complementary in providing critical information about the cancer genome.

Despite many ctDNA-related studies published in recent years, large-scale analysis of ctDNA across diverse cancer types is still lacking. To fill this gap and understand the Chinese-specific cancer genome, we employed a targeted deep sequencing assay that utilizes extensive error correction methods and provides the depth and breadth necessary to optimally investigate tumor-derived genomic alterations in plasma cfDNA, even at low allelic fractions (AFs)[16,17]. To eliminate the interference of clonal hematopoiesis (CH)-related variants, we also performed parallel sequencing for white blood cell (WBC) DNA and performed comparisons with cfDNA to identify tumor-specific genomic alterations, which we benchmarked against a tissue-based genomic database. Furthermore, we used ctDNA to quantify intratumor heterogeneity and clonality and explored the landscape of sensitive markers to targeted therapies in pan-cancer patients. The results of this study provide a foundation for further exploration and application of ctDNA and identify limitations in need of additional strategic and technological improvements.

## Results

**Description of analytical cohort**. We obtained 14,972 peripheral blood samples from 12,337 patients. DNA isolated from plasma and matched WBCs underwent hybridization capture and targeted deep sequencing to detect somatic single-nucleotide variants (SNVs), small fragment of insertions and deletions (Indels), copy number variants, and chromosomal rearrangements. Samples were excluded if they were from the same patient but had contradictory clinical records ($n = 309$), insufficient sequencing depth (<1000×, $n = 459$), abnormal contamination rate for cfDNA (>1%, $n = 670$), or an aberrant mismatch between cfDNA

and WBC DNA ($n = 201$). In total, 13,333 blood samples from 11,525 patients were included in our analysis (Fig. S1). General patient clinical information is summarized in Supplementary Data 2.

This cohort encompassed 41 principal tumor types. The most common tumor type was non-small cell lung cancer (NSCLC; $n = 5548$). Other common types included colorectal cancer ($n = 1195$), breast cancer ($n = 1178$), upper gastrointestinal (UGI) cancer ($n = 575$), and hepatocellular carcinoma (HCC; $n = 571$). The primary site was absent for 582 samples labeled as "unknown primary." Also, histologic classification was absent for 1668 lung cancer samples labeled as histology-unknown lung cancer. Over half of the samples were from patients with metastatic stage (clinical stage IV, 7303/13,333, 54.8%), 11.4% (1519/13,333) were from patients with localized or regional stages (stages I–III), and 33.8% (4511/13,333) lacked detailed staging information (Fig. S2). The average extracted concentration of cfDNA was 31.9 ng/ml (range, 1.05–396.0 ng/ml), with no apparent difference among the cancer types (Fig. S3). All plasma samples were sequenced to deep coverage (median, 4429×; range, 1000–30,368×) to ensure high sensitivity for detecting genomic alterations. The median sequencing depth for WBCs was 423× (range, 369–7279×).

**Some cfDNA mutations originate from CH variants in WBCs**. The CH variants identified in WBCs were traced in cfDNA, which highlighted 2754 mutations in 1861 plasma samples (14.0%) (Supplementary Data 3 and Fig. 1a). Fifteen canonical genes associated with CH, including *DNMT3A, TP53, TET2,* and *PPM1D*, were the most recurrent (Fig. 1a). The CH variants were detected in >20% of plasma samples from melanoma, bladder, uterine, and prostate cancers and in <10% of samples from renal and thyroid cancers. The top detectable CH genes varied across cancer types. For instance, *TP53* was the most frequent CH gene in melanoma and small cell lung cancer (SCLC), whereas *DNMT3A* was most frequent in uterine and cervical cancers. When we interrogated the AF distribution of CH variants in cfDNA, most were between 0 and 10% (Fig. 1b). Furthermore, CH variants exhibited significantly lower AFs than non-CH mutations in cfDNA ($p < 0.001$; Supplementary Data 3 and Fig. 1c). Previous reports indicated that circulating CH variants tend to emerge with age[18,19], and consistent with our observations, CH variants were found in 9.1% of patients aged <40 years and in 23.1% of patients aged >80 years (Fig. 1d).

Although this mutation filter strategy largely avoided interference by variants with low AF in cfDNA, the limited sequencing depth of WBCs raises the concern that some CH variants could escape WBC screening and thus bias the overall cfDNA findings. To address this issue and clarify the biological sources of somatic mutations in cfDNA, we collected paired tumor tissue samples from 1291 patients and conducted the same sequencing procedures. The median sequencing depth for tissue samples was 1808× (range, 487–5525×). In plasma cfDNA from the same patients, 4500 somatic mutations were detected in 976 samples (75.6%). Based on their detectability in different samples, we classified these variants into three categories: 316 WBC-matched variants (co-occurring in both plasma and WBC, equal to CH variants), 4023 biopsy-matched variants (co-occurring in both plasma and tissue biopsy), and 161 variants of an unknown source (VUSOs; only occurring in plasma) (Supplementary Data 6). At an individual level, 941 plasma samples (72.8%) harbored biopsy-matched variants, 214 plasma samples (16.6%) harbored WBC-matched variants, and 124 plasma samples (9.6%) harbored VUSOs (Fig. 1e). These results suggested that the vast majority of cfDNA somatic mutations filtered by WBCs (96.2%, 4032/4193) could be verified in matched tumor tissue. Totally 10

(*DNMT3A, TP53, TET2, ASXL1, PPM1D, ATM, JAK2, SF3B1, CHEK2, CBL*) out of 15 canonical CH-related genes (see "Methods") were involved in either WBC-matched variants or VUSOs. Of the 316 WBC-matched variants, 27.5% were in the 10 canonical CH-related genes, while of the 161 VUSOs a smaller

proportion (14.9%, Chi-square $p = 0.002$) were in these CH-related genes (Fig. 1f). The AFs were similar between the two aforementioned components and significantly lower than that of the biopsy-matched variants (Fig. 1g). It seemed that CH variants might still be present even after WBC-matched screening.

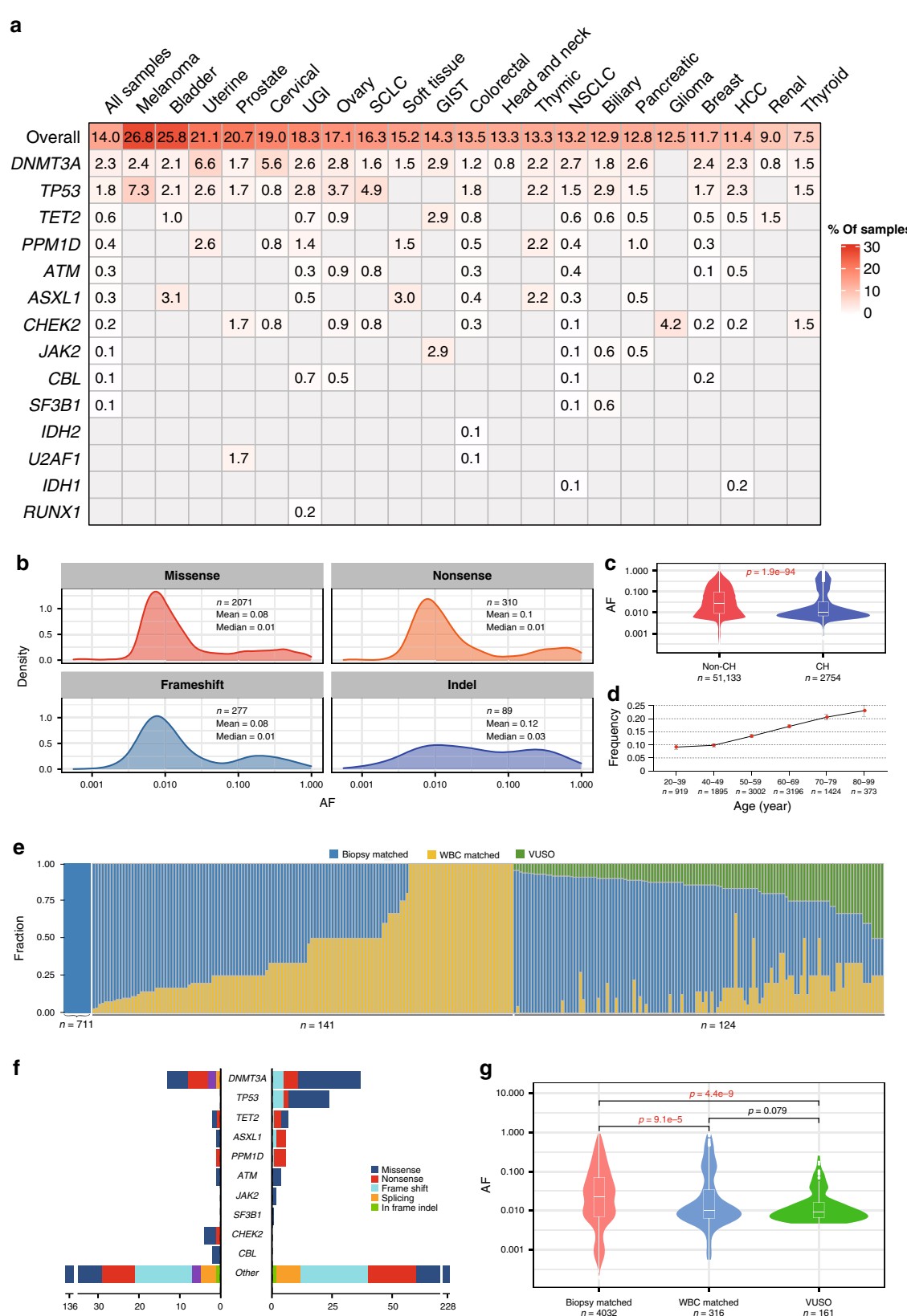

**Fig. 1 Identifying CH variants in plasma cfDNA via matched WBC sequencing. a** Percentage of plasma samples with identified CH variants in different cancer types. The first row indicates the overall percentage of samples with any CH variants in different cancer types, and the other rows indicate the percentage of samples with CH variants in 15 canonical genes. Gray nubs indicate that no CH variants were detected. **b** Density distribution of AFs of identified CH variants in cfDNA. Different panels indicate the distribution of different mutational types. **c** AFs of CH variants in cfDNA were significantly lower than non-CH mutations (two-sided Mann–Whitney $U$ test). Centre line, median; box limits, upper and lower quartiles; whiskers, data range. **d** The percentage of plasma samples with CH variants increases with the age of patients. Error bars indicate SEM. **e** Relative fractions of biopsy-matched variants, WBC-matched variants, and VUSOs in each plasma cfDNA with matched tumor tissue and WBC sequencing. **f** Distribution of WBC-matched variants and VUSOs according to gene categories. Only canonical CH-related genes are shown. **g** Comparison of variant AFs among biopsy-matched variants, WBC-matched variants, and VUSOs (two-sided Mann–Whitney $U$ test). Centre line, median; box limits, upper and lower quartiles; whiskers, data range. AF allele frequency, CH clonal hematopoiesis, VUSO variant of unknown source, WBC white blood cell, GIST gastrointestinal stromal tumor, HCC hepatocellular carcinoma, NSCLC non-small cell lung cancer, SCLC small cell lung cancer, UGI upper gastrointestinal cancer.

However, the proportion would be limited compared with tumor-derived variants. Subclonal alterations missed in single-region sequencing or other (non-WBC) source of background noise might constitute the remaining proportion of VUSOs.

**Detectability and molecular features of ctDNA vary across different cancer types.** After elimination of CH variants, 51,133 nonsynonymous mutations (Fig. S4), 1945 copy number variants (Supplementary Data 4), and 442 rearrangements (Supplementary Data 5) were detected in 9801 plasma samples. The overall sensitivity of ctDNA detection was 73.5% (Fig. S5). The detectability of ctDNA varied across different cancer types, with SCLC (91.1%, 95% confidence interval [CI] 88.5–93.7%), prostate cancer (87.9%, 95% CI 83.6–92.2%), uterine cancer (77.6%, 95% CI 72.8–82.4%), HCC (77.1%, 95% CI 75.3–78.9%), and UGI cancer (76.9%, 95% CI 75.1–78.7%) showing a higher sensitivity compared with other cancer types. A low ctDNA detectability was observed for thyroid (41.8%, 95% CI 35.8–47.8%) and renal (56.4%, 95% CI 52.1–60.7%) cancers (Figs. 2a and S5).

Most stage IV samples (79.7%, 5817/7303) had detectable alterations, whereas a smaller proportion of stage I–III samples (57.9%, 879/1519) had detectable alterations (Fig. S5). Therefore, we interrogated the detection of ctDNA for various cancer types at different clinical stages. The sensitivity of ctDNA detection in stage I–III disease was >60% for SCLC (87.5%, 95% CI 75.8–99.2%), HCC (68.0%, 95% CI 62.6–73.4%), NSCLC (63.4%, 95% CI 61.6–65.2%), and cervical cancer (60.9%, 95% CI 50.7–71.1%) (Fig. S5). In stage IV disease, ctDNA was detected in >70% of samples for most cancer types except for renal cancer (63.2%, 95% CI 57.4–69.0%) and melanoma (68.2%, 95% CI 58.3–78.1%).

The maximal AF of ctDNA, which represents the relative quantification of tumor-derived ctDNA molecules within the total population of cfDNA molecules, is correlated with the cellular amount of tumor mass and can be utilized to determine tumor load[15]. We found that ctDNA AF varied dramatically across different cancer types and different samples of the same cancer type (Fig. 2b). SCLC exhibited a high AF compared with other types (versus prostate cancer, $p < 0.001$; versus NSCLC, $p < 0.001$), whereas the lowest AF was observed for cerebral glioma (versus thyroid cancer, $p = 0.0247$), possibly due to the blood–brain barrier.

The blood tumor mutational burden (bTMB) has been introduced as a promising predictive biomarker for immunotherapy[20,21]. In addition to SCLC demonstrating a high bTMB (versus uterine cancer, $p < 0.001$, Fig. 2c), uterine, bladder, colorectal, and cervical cancer also had a higher bTMB than other cancer types. The lowest bTMB was observed for thyroid cancer and glioma. The bTMB ranking within this cohort was similar to that in MSKCC[2], but there were some discrepancies; melanoma and glioma showed higher tissue TMB rankings in MSKCC but lower bTMB rankings in our cohort, suggesting that ctDNA

shedding might affect the determination of circulating tumor burden. Based on the 75% percentile of the bTMB (Fig. 2d), the cut-off for a high and low bTMB was set as 8.7 mutations/Mb. Over half of the SCLC samples were deemed as high bTMB (65/114, 57.0%, Fig. 2e). In addition, NSCLC (1411/4243, 33.3%), bladder cancer (23/74, 31.1%), colorectal cancer (256/878, 29.2%), and cervical cancer (25/97, 25.8%) also exhibited larger fractions of high bTMB samples, indicating that the cut-off value of the bTMB should be separately determined for different cancer types. Furthermore, the bTMB was positively correlated with the ctDNA AF ($R^2 = 0.106$, $p < 0.001$), and a ladder-like increase in the bTMB was observed when we divided samples into different groups according to the ctDNA AF (Fig. S6). These results suggested that the AF-corrected bTMB may be a better therapeutic marker, although this should be validated in specifically designed studies.

Because the large ranges of the AF and bTMB imply genomic heterogeneity, we quantified tumor heterogeneity using the mutant-allele tumor heterogeneity (MATH) algorithm. Most cancer types showed significantly higher MATH values than corresponding types in MSKCC (Fig. 2f). This indicated the potential advantage of ctDNA over single-region biopsy in reflecting tumor heterogeneity, which could be because a single-region biopsy tends to omit some subclonal alterations with the inferior AF. Together, these results highlighted distinct ctDNA characteristics among different cancer types and in comparison with tissue sequencing data.

**Mutational landscape of pan-cancer ctDNA.** We further investigated the specific mutational landscape of pan-cancer ctDNA. Half of the plasma samples (50.5%) harbored *TP53* mutations, and the frequencies of *TP53* mutations were <20% in cervical cancer, gastrointestinal stromal tumors (GIST), thyroid cancer, and melanoma (Fig. 3a). Other common mutant genes in different cancer types were as expected, based on previous knowledge. For instance, *EGFR* mutations occurred in 44.2% of NSCLC samples, and *RB1* mutations occurred in 63.4% of SCLC samples. Besides *TP53*, *APC* (44.4%), *KRAS* (33.8%), and *PIK3CA* (11.2%) were the most common mutant genes in colorectal cancer samples. *TERT* promoter SNVs (22.3%) were commonly found in HCC samples. Pancreatic cancer samples exhibited high frequencies of *KRAS* (62.8%) and *CDKN2A* (17.2%) mutations. Also, 39.1% of breast cancer samples harbored *PIK3CA* mutations. Remarkably, *DNMT3A* mutations, of which a considerable portion were filtered out as CH variants, generally occurred across all the cancer types but were absent in MSKCC. This indicates that additional techniques, other than WBC filtration, should be performed to further eliminate the interference of CH variants.

A previous study provided information about putative driver genes for different cancer types[22]. To evaluate genomic concordance between ctDNA-based liquid biopsy and tissue biopsy, we performed correlation analysis of the frequencies of

putative driver genes, which showed a strong linear relationship ($R^2 = 0.87$, $p < 0.001$, Fig. 3b). Despite this overall concordance, there were some disharmonies between the mutational landscape of our cohort and MSKCC. To elucidate these disharmonies, the frequencies of top mutant genes were further compared with those in MSKCC; only those with a statistically significant

difference and >50%-fold change were identified as differential genes, to avoid differences due to sample size (Fig. S7). *TP53* mutations were more common in HCC plasma samples than in MSKCC (53.1% versus 28.6%), and R249S, which is closely associated with aflatoxin exposure and hepatitis B virus infection[23,24], was enriched in plasma ctDNA (Figs. S7 and

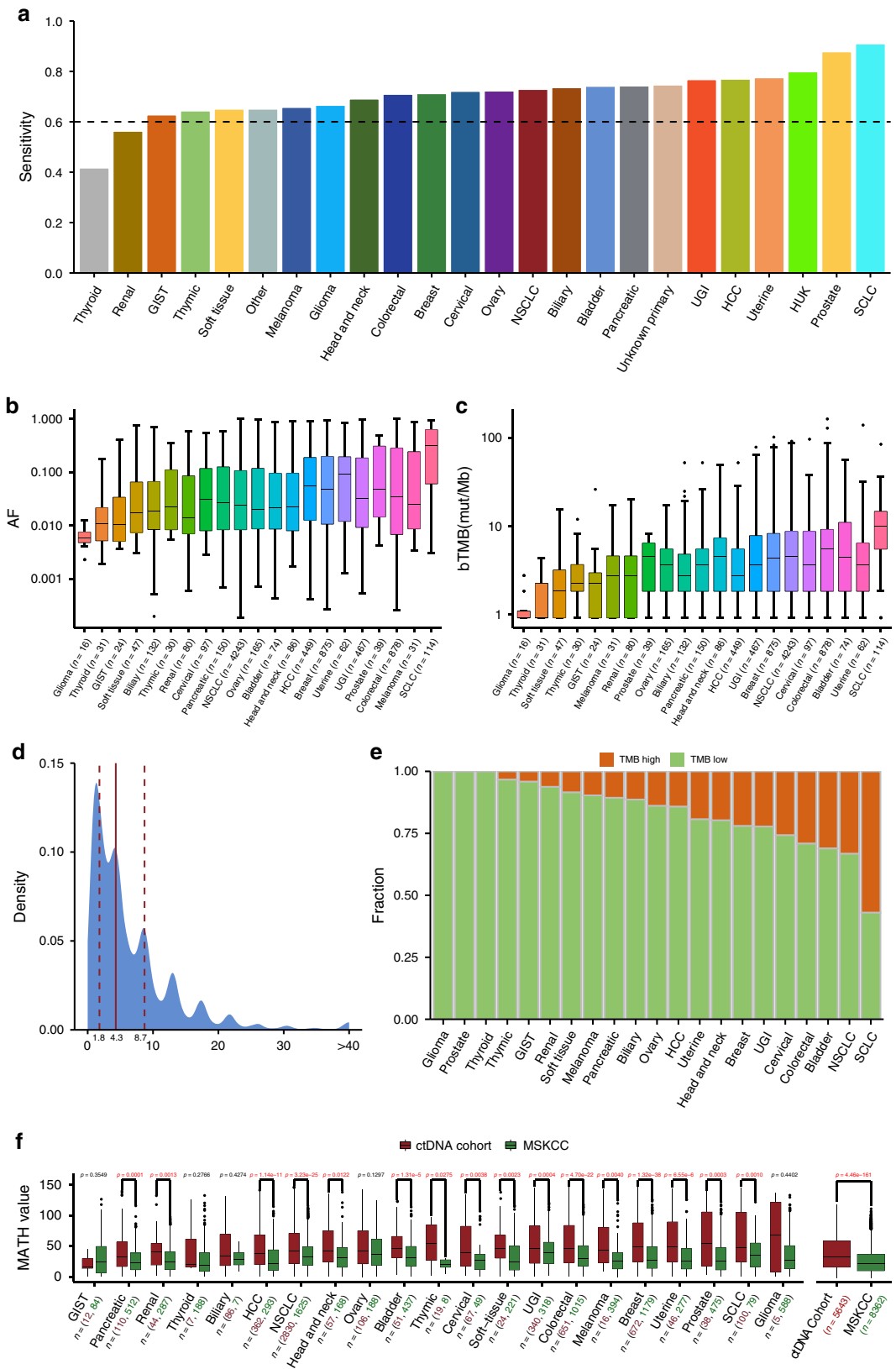

**Fig. 2 Detectability of ctDNA in pan-cancer plasma. a** Detection sensitivity of ctDNA in multiple cancer types. **b** AFs of ctDNA mutations varied across different cancer types. Median values are represented by black lines within the bars. For samples with multiple mutations, the highest AF is highlighted. Centre line, median; box limits, upper and lower quartiles; whiskers, 1.5× interquartile range; points, outliers. **c** Different cancer types showed different bTMB. Median values are represented by black lines within the bars. Centre line, median; box limits, upper and lower quartiles; whiskers, 1.5× interquartile range; points, outliers. **d** Density distribution of bTMB of all enrolled samples. Vertical solid and broken lines indicate the median and upper/lower quartile values, respectively. Based on the upper quartile of bTMB, the cut-off for high and low bTMB was set as 8.7 mutations/Mb. **e** Relative fractions of high and low bTMB samples in different cancer types. **f** Comparison of MATH values for different cancer types between our ctDNA and MSKCC cohorts (two-sided Mann–Whitney $U$ test). The bottom red and green numbers indicate the sample sizes of corresponding tumor subtypes in the ctDNA cohort and MSKCC, respectively. Median values are represented by black lines within the bars. Centre line, median; box limits, upper and lower quartiles; whiskers, 1.5× interquartile range; points, outliers. AF allele frequency, bTMB blood tumor mutational burden, GIST gastrointestinal stromal tumor, HCC hepatocellular carcinoma, NSCLC non-small cell lung cancer, SCLC small cell lung cancer, UGI upper gastrointestinal cancer.

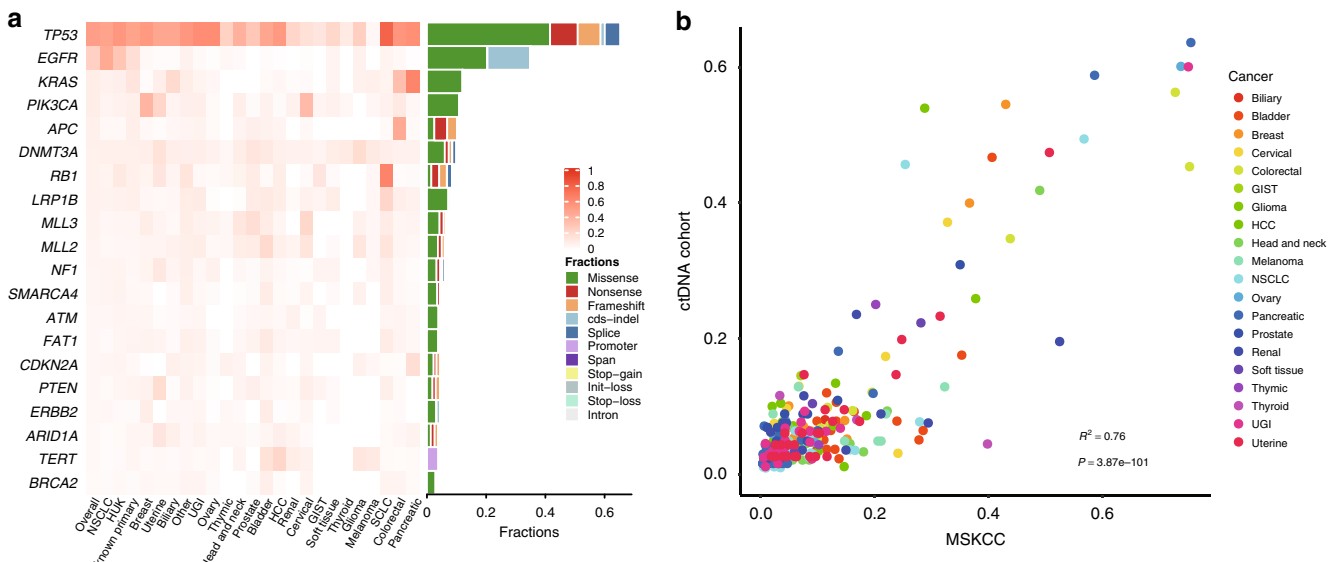

**Fig. 3 Mutational landscape revealed by ctDNA profiling. a** Heatmap illustrating the top 20 most common mutant genes. Color gradation represents the mutational prevalence of each gene in different cancer types. The right bars indicate the distribution of different mutational types for each displayed gene. **b** Mutational prevalence of driver genes in our ctDNA and MSKCC cohorts. Putative driver genes in different cancer types are displayed. Symbol colors correspond to different cancer types. The statistical test used is two-tailed Pearson correlation test. GIST gastrointestinal stromal tumor, HCC hepatocellular carcinoma, NSCLC non-small cell lung cancer, SCLC small cell lung cancer, UGI upper gastrointestinal cancer.

S8A). *EGFR* mutations were more common in NSCLC plasma samples than in MSKCC (44.8% versus 25.2%, Fig. S7), and the distribution of hot spots showed some discrepancies. For example, the frequencies of exon 19 deletion (e19del) and T790M were higher in plasma samples, and C797S, which mediates resistance to third-generation epidermal growth factor receptor (EGFR) inhibitors[25], existed only in plasma samples (Fig. S8B). A lower *KRAS* mutation frequency was observed in NSCLC plasma samples than in MSKCC (6.8% versus 27.2%) (Fig. S7). Notably, a high frequency of *AR* mutation was observed in prostate cancer plasma samples (Fig. S7). A detailed distribution of the mutation sites revealed that W742C and T878A were most common in plasma samples, whereas L702H and H875Y were most common in MSKCC (Fig. S8C). All of these mutational sites were located at the ligand-binding domain and are related to resistance to androgen-deprivation therapy[26]. These findings underline the advantages of ctDNA for uncovering drug resistance mechanisms and highlight similarities and discrepancies between plasma and tissue biopsy. One possible reason for these discrepancies are racial differences. That is, the distinct mutational prevalence, observed in the present study, may implicate the pathogenetic driving features of cancer in Chinese patients. However, several other factors might also contribute to these discrepancies, including the sample source, sequencing panel, detection sensitivity, patient gender and age

distributions, and other demographic variables, which should be further clarified using strict experimental controls.

To explore the functional implication of ctDNA variants, 10 canonical signaling pathways (i.e., cell cycle, Hippo, Myc, Notch, Nrf2, PI3K, RTK/RAS/MAPK, TGFβ, p53, and Wnt) associated with proliferative potential were evaluated by allocating specific genes to each pathway[27]. The RTK-RAS pathway showed the highest frequency of alterations (56.1% of samples), followed by the p53 (53.9% of samples) and PI3K (22.8% of samples) pathways (Fig. 4). We further investigated the mutual exclusivity and co-occurrence of alterations among the different pathways. The most salient feature was that there was no co-occurrence between the RTK-RAS pathway and other pathways, and a slight mutual exclusivity was found between the RTK-RAS and p53/PI3K pathways, suggesting that alterations in RTK-RAS pathway alone could induce oncogenesis (Fig. S9). The strongest co-occurrence was found between p53 and cell cycle pathways, and some rarely altered pathways, such as Notch, Wnt, and TGF-beta pathways, were usually co-altered (Fig. S9). These results illustrate the cross-talk among the different pathways, which reveals functional interactions and dependencies that could be therapeutically explored.

**Rebuilding tumor clonal structure via ctDNA profiling.** The high heterogeneity of ctDNA reflected by the MATH algorithm

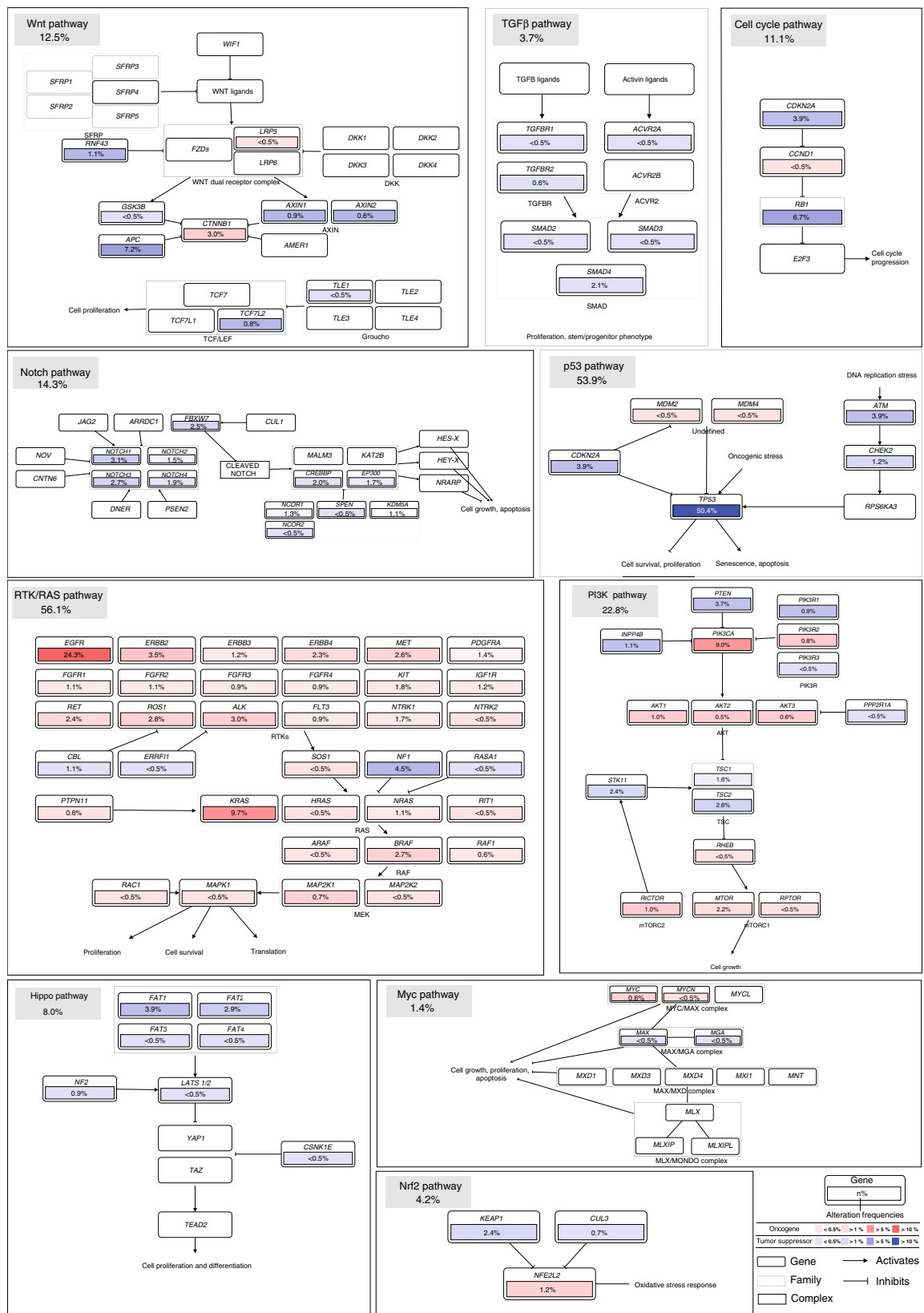

**Fig. 4 Pathway members and interactions in the ten selected pathways.** Oncogenes and tumor-suppressor genes are illustrated with red and blue, respectively. Color intensity indicates the frequency of alteration within the entire dataset. Blank boxes represent genes not covered in our sequencing panel.

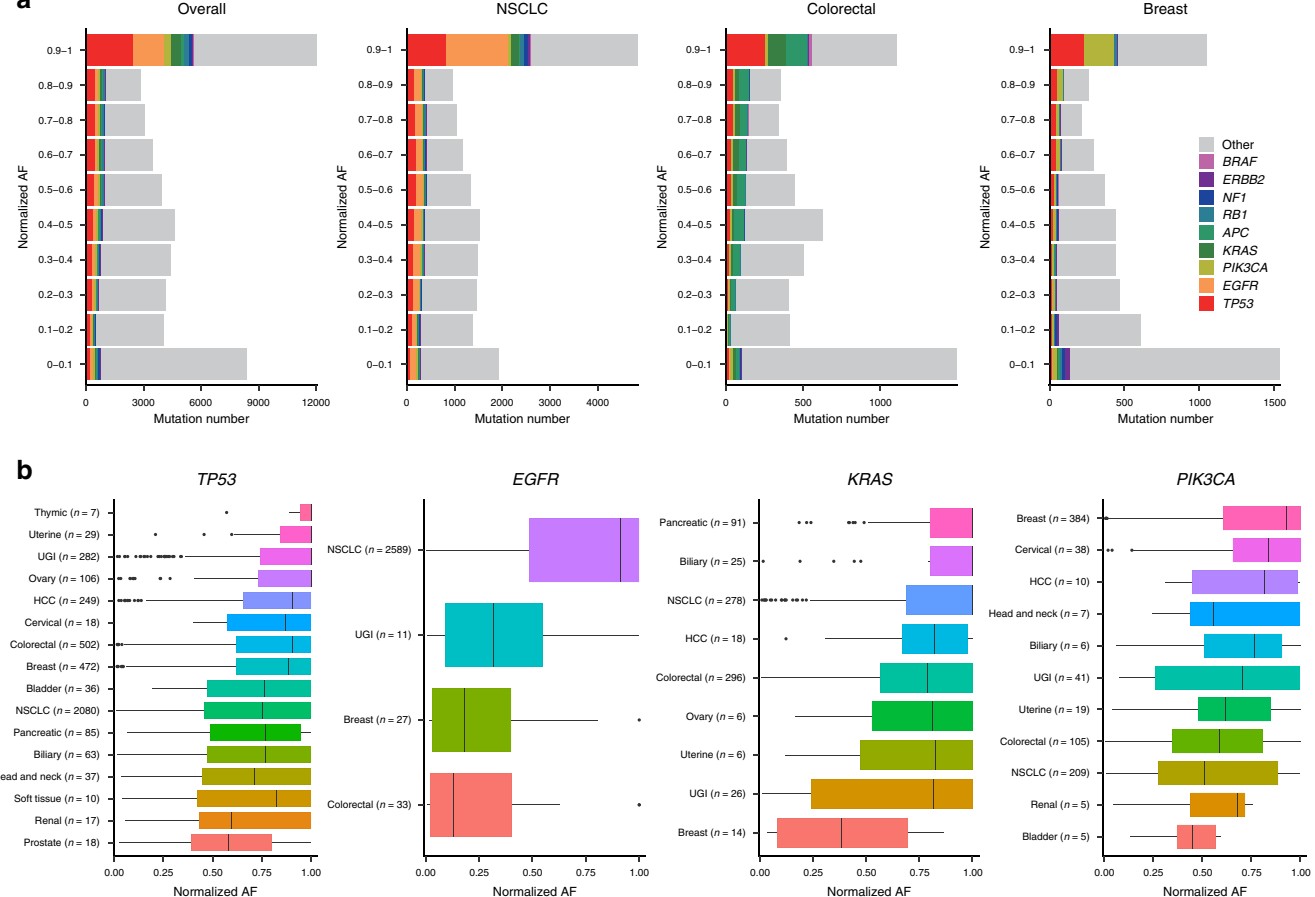

**Fig. 5 Estimated ctDNA clonality is consistent with genomic characteristics of cancer subtypes. a** Number of mutations with different ranges of clonality in the overall cohort, NSCLC, colorectal cancer, and breast cancer. Several common driver genes are highlighted with different colors and demonstrate shifts in fractions within different clonality ranges. **b** Clonality distribution of *TP53*, *EGFR*, *KRAS*, and *PIK3CA* in different cancer types. In each panel, cancer types with fewer than five specific mutations (*TP53*, *EGFR*, *KRAS*, and *PIK3CA*) are not shown. Centre line, median; box limits, upper and lower quartiles; whiskers, 1.5× interquartile range; points, outliers.

implies its complicated clonal structure. Considering that most samples were from patients with an advanced disease stage who were receiving treatment, the abundance of ctDNA enabled the rebuilding of genomic clonality. We employed a method for quantifying mutational clonality using normalized AFs and explored their distribution across different cancer types and specific driver genes.

In the overall plasma cohort, the mutational clonality distribution was nearly bimodal, with two maxima typically falling >0.9 and <0.1, respectively (Fig. 5a). The percentage of 9 common driver genes (i.e., *TP53*, *EGFR*, *PIK3CA*, *KRAS*, *APC*, *RB1*, *NF1*, *ERBB2*, and *BRAF*) was 47.1% in the subgroup with a clonality ≥0.9 but gradually declined to 9.5% in the subgroup with a clonality <0.1. A similar pattern persisted when focusing on NSCLC, colorectal cancer, or breast cancer samples, but the dominating clonal driver genes, except for *TP53*, differed among these three types: *EGFR* (28.2%) in NSCLC, *APC* (12.2%) and *KRAS* (10.4%) in colorectal cancer, and *PIK3CA* (18.5%) in breast cancer (Fig. 5a). *TP53* was the most prevalent mutant gene across samples, with an average clonality >0.5 among the different cancer types, with high values for thymic and uterine cancer (Fig. 5b). The clonality of *EGFR* mutations was high in NSCLC. A considerable fraction of *EGFR* mutations, particularly for UGI, breast, and colorectal cancer, were subclonal variants that were mainly located at the extracellular domain of the coding protein, probably induced by anti-HER2 therapy. For *KRAS* mutations,

the highest clonality was present in pancreatic and biliary cancer, whereas the lowest clonality was present in breast cancer. *PIK3CA* was frequently altered in various cancer types, with the highest clonality present in breast and cervical cancer. These results are consistent with prior knowledge and suggest that ctDNA-directed clonality analysis is a favorable alternative to determine the tumor clonal structure.

**Therapeutic actionability revealed by ctDNA profiling.** One of the most meaningful applications of ctDNA-based liquid biopsy is to seek drug-sensitive markers and identify resistant mechanisms for patients who continue to progress on targeted therapies. Thus we used our large cohort to systematically evaluate therapeutic actionability as revealed by ctDNA profiling. We analyzed 8032 plasma samples with genomic alterations and excluded samples of histology-unknown lung cancer and unknown primary cancer. Based on criteria established by the OncoKB database, and other evidential reports, we globally detected 4665 potential drug-sensitive targets from 3306 samples (41.2%) (Supplementary Data 7), of which 2299 samples (28.6%) harbored Level 1 targets, 100 (1.2%) harbored the highest Level 2A targets, 221 (2.8%) harbored the highest Level 2B targets, 226 (2.8%) harbored the highest Level 3A targets, and 459 (5.7%) harbored the highest Level 3B targets (Fig. S10). Over half of NSCLC (55.8%, 2258/4050), GIST (54.5%, 12/22), and breast cancer

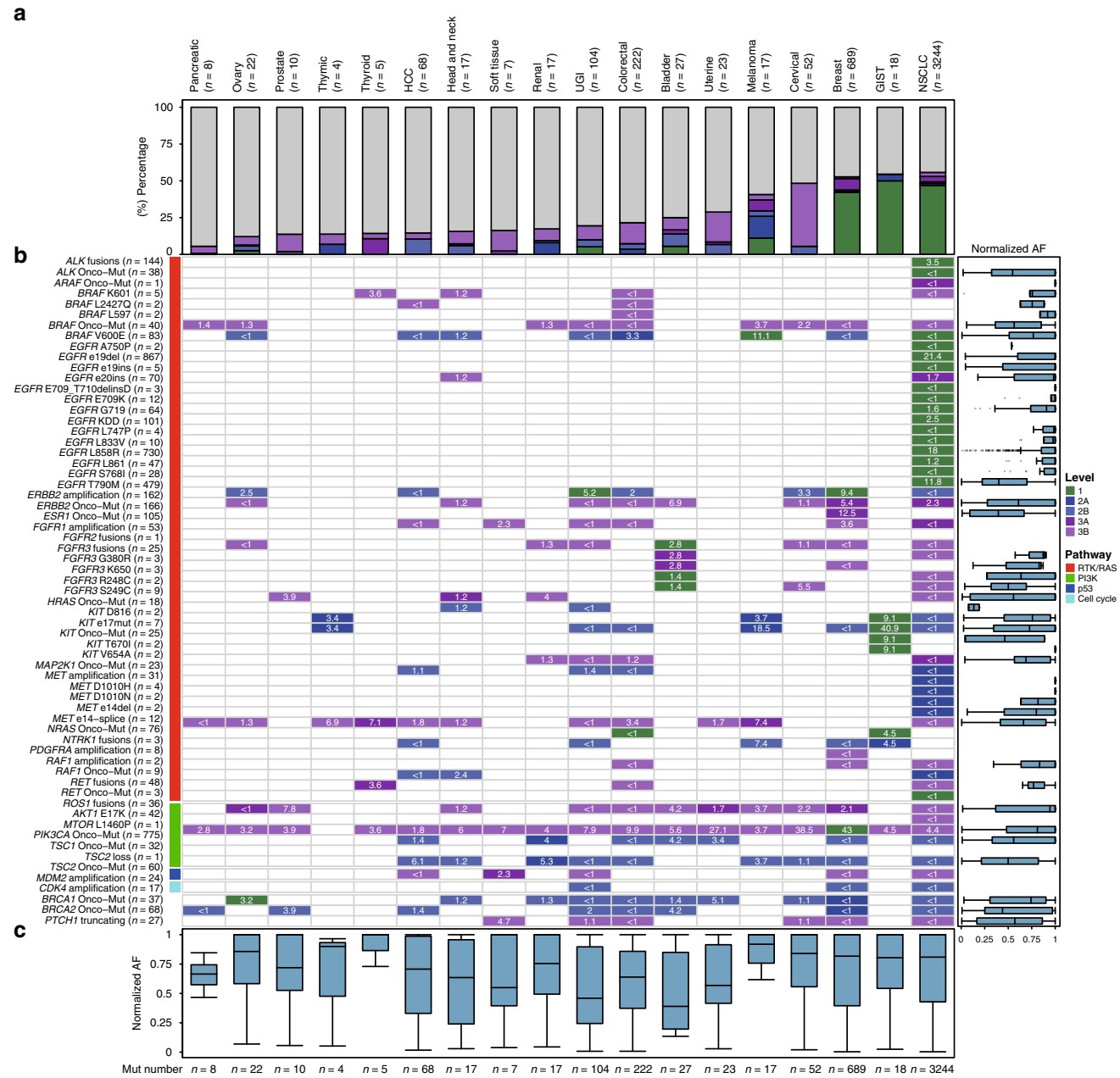

**Fig. 6 Overview of therapeutic actionability revealed by ctDNA profiling. a** Frequencies of clinical actionability across different cancer types, broken down by evidence levels. GIST gastrointestinal stromal tumor, HCC hepatocellular carcinoma, NSCLC non-small cell lung cancer, SCLC small cell lung cancer, UGI upper gastrointestinal cancer. **b** Frequencies of actionable alterations across cancer types. Alterations are grouped by pathway. The right box chart indicates the clonality distribution of different actionable alterations. Centre line, median; box limits, upper and lower quartiles; whiskers, 1.5× interquartile range; points, outliers. **c** Clonality distribution of actionable alterations in different cancer types. Centre line, median; box limits, upper and lower quartiles; whiskers, data range.

(52.7%, 443/840) samples exhibited therapeutic markers. Level 1 markers were observed in NSCLC, breast cancer, UGI cancer, GIST, ovary cancer, bladder cancer, melanoma, and colorectal cancer (Fig. 6a). Of note, actionable variants were absent only in cerebral glioma samples, implying genomic specificity and underlining the poor therapeutic potential for central nervous system neoplasms.

In total, 63 types of drug-sensitive markers were detected, the most frequent of which was *EGFR* e19del (Supplementary Data 7 and Fig. 6b). *PIK3CA* mutations were detected in up to 17 cancer types. In NSCLC, the dominant mutations were *EGFR* e19del and L858R, and the clonality of T790M was lower than

that of e19del and L858R. This indicated the evolutionary trajectory of acquired resistance for patients using generation I and II tyrosine kinase inhibitors. Although ctDNA detectability was poor in GIST, we frequently detected drug-sensitive markers represented by *KIT* mutations targeted by regorafenib, imatinib, and sunitinib. In several cancer types with the highest likelihood of harboring drug-sensitive markers (i.e., NSCLC, GIST, breast cancer, cervical cancer, and melanoma), the clonality of markers was typically higher than that of other types, indicating that their cellular fraction of drug-sensitive clones was high and thus suggesting the potential efficiency and efficacy of targeted agents (Fig. 6c).

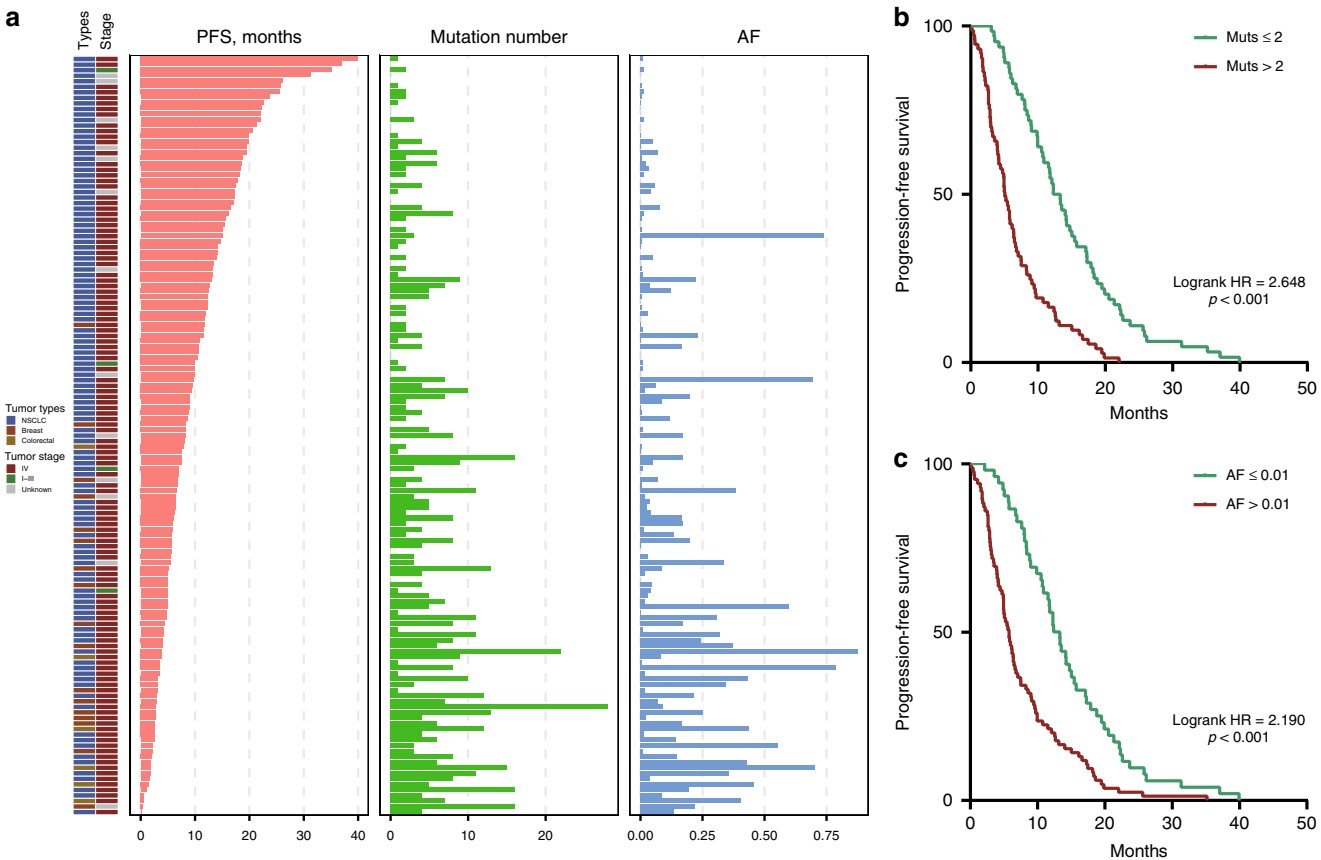

**Fig. 7 Circulating tumor burden determined by ctDNA profiling is associated with therapeutic prognosis. a** Overview of tumor types, clinical stages, PFS, ctDNA mutation numbers, and ctDNA AFs for 137 patients receiving targeted therapies guided by ctDNA profiling. **b** Patients with >2 mutations in ctDNA exhibited poorer PFS than those with ≤2 mutations in ctDNA. **c** Patients with >0.01 ctDNA AF exhibited poorer PFS than those with ≤0.01 ctDNA AF.

Clinical follow-up was successfully completed for 137 patients receiving targeted therapies in accordance with ctDNA profiling, including 114 NSCLC, 17 breast cancer, and 6 colorectal cancer patients (Fig. 7a and Supplementary Data 8). The vast majority of patients were stage IV, and progression-free survival (PFS) ranged from 4 to 1198 days. Both mutation number and the AF of plasma ctDNA exhibited moderately negative correlations with patient PFS (Figs. 7a and S11). The median PFS for patients with ≤2 mutations was 12.8 months, which was significantly longer than that for patients with >2 mutations (median, 5.1 months; log-rank hazard ratio = 2.648, 95% CI 1.838–3.815, $p < 0.001$; Fig. 7b). Likewise, patients with a ≤0.01 AF (median, 12.8 months) showed significantly longer PFS than those with a >0.01 AF (median, 5.8 months; log-rank hazard ratio = 2.190, 95% CI 1.561–3.073, $p < 0.001$; Fig. 7c). These results suggest that ctDNA could also help to predict therapeutic prognosis. For patients with a high circulating tumor burden, therapeutic efficacy may be impeded by genomic heterogeneity, and subclonal expansion might lead to tumor progression. Therefore, by integrating pathway, clinical actionability, and clonality analysis, we were able to assess the potential benefits of targeted agents that could be further explored in experimental and clinical contexts.

## Discussion

Herein we report a study of noninvasive ctDNA detection for pan-cancer Chinese patients. We developed a parallel sequencing process to identify CH-associated variants, interrogated the sensitivity of ctDNA, and described the fluctuant level and bTMB distribution across different cancer types. We found that the genomic landscape of ctDNA differed from that of tissue sequencing data. Furthermore, through liquid biopsy, we rebuilt the tumor clonal structure and highlighted genomic alterations with clinical actionability, which helps gauge the potential value of future lines of therapeutic development.

A remarkable feature of this study is the inclusion of patients with early-stage disease. Currently, the most promising application of liquid biopsy is cancer screening and early detection. As the most popular auxiliary diagnostic approach, protein biomarkers (e.g., cancer antigen 125, carcinoembryonic antigen, prostate-specific antigen, and cancer antigen 19-9) are usually restricted to one or several cancer types and have poor specificity[28–32]. The development of liquid biopsy allowed for new insights into the early detection of multiple solid tumors but faces serious challenges regarding specificity and sensitivity[33]. In this study, the sensitivity of ctDNA was 57.9% for localized or regional disease. Although this is still far from ideal, this is superior to the sensitivity of common protein biomarkers. Furthermore, according to previous reports[34], the specificity of ctDNA in healthy control individuals is >95%, further confirming the feasibility of using ctDNA for cancer screening and early detection. Several optimizations such as the integration of methylated modifications[35], cfDNA fragment information[36], and other clinical parameters can further improve detection sensitivity based on liquid biopsy.

In general, the patterns and frequencies of the main driver genes presented in this study were similar to those revealed by large-scale tissue sequencing data, which confirms the credibility and feasibility of using ctDNA profiling to guide precision medicine practice. However, some confounding factors still

impede ctDNA determination and should be clarified. CH variants emerge with age and are identified as a main confounding factor in ctDNA-related analysis[37]. Although WBC- and tissue-matched sequencing successfully identified a considerable amount of CH variants and verified their tumor origin for the vast majority of cfDNA mutations, some highly suspicious mutations such as DNMT3A and TET2[18,19] were still retained. Such data indicates that other, non-WBC sources of background noise can determine tumor-derived ctDNA. Razavi et al.[38] recently performed high-depth sequencing to reveal sources of cfDNA mutations. Even with >60,000× raw depth, a fraction of the cfDNA mutations, including 15 canonical CH genes, were missed in WBC-matched sequencing, which is in line with our findings. Thus an expanded cfDNA sequencing database encompassing both healthy individuals and cancer patients could further reduce interference from CH variants and non-CH noise. Also, discrepancies between our and MSKCC cohorts could also be due to differences in the sequencing panel, detection sensitivity, patient gender and age distributions, and racial origin. For instance, a meta-analysis demonstrates that EGFR mutation prevalence is associated with ethnicity and smoking history. The prevalence of EGFR mutations was higher in Asian women than in Caucasian/mixed ethnicity women, and a greater smoking history was associated with a decreased odds of exhibiting the EGFR mutation, particularly for patients with >30 pack-years[39]. Considering the relatively high incidence of NSCLC in China compared with developed countries, this finding may serve as another reminder to prioritize smoking control in China.

Intratumor heterogeneity poses a major problem for tumor management, and tumor clonal structure is critical for treatment determination. For instance, Lee et al. performed genomic profiling across 127 multisector or longitudinal specimens from glioblastoma patients. Chemical screening of patient-derived glioma cells showed that the therapeutic response was associated with genetic similarity, and targeting clonal trunk events was more effective in reducing tumor burden than targeting subclonal events[40]. Also, emerging evidence shows that clonal TMB and predicted neoantigen burden are closely associated with survival and immunotherapy efficacy[41,42]. Using the method developed by Zill et al.[43] and used in the present study, clonal structure can be determined despite tumor heterogeneity and the tumor impurity introduced by single-region tissue sampling. Thus this technique could be employed to develop optimized biomarkers associated with patient survival and therapeutic efficacy, such as the clonal bTMB and clonal blood neoantigen burden. In addition, conventional pharmaceutical small-molecule approaches, such as EGFR tyrosine kinase inhibitors that target single oncogenic drivers, could also benefit from ctDNA-directed clonal analysis. For example, patients harboring clonal EGFR-sensitive mutations may exhibit better outcomes than those with subclonal mutations.

Recently, two prospective studies using plasma-based genotyping to guide targeted therapy in NSCLC provided support for the potential incorporation of plasma next-generation sequencing into practice guidelines[44,45]. One of these studies demonstrated a disease control rate of up to 85.7% under the guidance of ctDNA profiling[45]. The results of the present study further expand the clinical actionability of ctDNA to other solid tumors, including some that were nearly absent in previous ctDNA-associated analyses, such as GIST. Overall, we found that 41.2% of patients may benefit from targeted therapy. Even for cancer types with few approved targeted agents (i.e., pancreatic cancer), our findings indicate that a fraction of patients could potentially benefit from targeted therapy as well. These results provide a foundation for future umbrella trials utilizing molecular genotyping to guide clinical decision-making for pan-cancer patients. Our study also

highlights the capacity of ctDNA to rebuild tumor clonal structure. By integrating clonality and actionability analyses, we provide a quantitative depiction of the relative abundance of tumor cells with drug-sensitive alterations, which can enable more precise clinical decisions than the simple qualitative detection of actionable alterations, as described in our previous work[46].

This "real-world" study also has some limitations. A primary limitation is the small number of patients with localized and regional disease, which is a result of multiple factors. First, most cancer patients in China are initially diagnosed with advanced stages of disease due to the poor development and prevalence of screening and early detection approaches. Second, the application of genomic analysis in the management of early-stage cancer is still controversial, which results in early-stage patients being less willing to undergo genomic testing than advanced-stage patients. Another limitation is the moderate sequencing depth due to the uneven quantity of blood sampling. In addition, most patients had incomplete clinical information, which restricted the joint analysis to a great extent. To solve these problems, close cooperation among the medical treatment alliance and scientific research institutions, as well as genomic testing enterprises, should be established and complete data sharing should also be implemented.

Despite these limitations, ctDNA-based liquid biopsy provides opportunities to more fully understand the cancer genome. This approach can also help guide clinical practice via the determination of the clonal structure and actionable mutations. Future work will include validation studies using an in-house established tissue-based cohort and prospective studies on the clinical application of ctDNA for multi-omics early detection systems and the delivery of personalized therapy. Our findings could also serve as a foundation for the development of new therapeutic targets and clinical trials using combined treatment strategies.

## Methods

**Patient characteristics, sample collection, and clinical follow-up**. Between January 2017 to July 2019, a total of 12,337 patients with >40 different cancer types were enrolled at Geneplus Medical Laboratory (Beijing, China). Tumor types were annotated according to an institutional classification system, OncoTree (http://www.cbioportal.org/oncotree/). This was an observational, non-interventional study and was approved by the ethical committee at Chinese PLA General Hospital. General patient characteristics (i.e., identification number, diagnostic age, gender) were collected during initial enrollment. Clinical stage was identified by experienced clinicians and re-checked during the data analysis process according to the American Joint Committee on Cancer TNM staging system for each cancer type; clinical stage was labeled as unknown for some patients due to contradictory records or dropouts. Treatment histories were collected from most patients.

During the initial enrollment period, at least 10 ml of peripheral blood was collected in cfDNA blood collection tubes (Streck, Omaha, NE, USA) at room temperature. Multiple blood collections were performed for certain patients throughout their therapeutic courses. To reveal sources of ctDNA variants, tumor tissue samples were obtained from 1291 patients via aspiration or surgical biopsy. To evaluate the correlation between ctDNA profiling and patient prognosis, clinical follow-up was performed for a small portion of patients, for whom two further selection criteria were used: (1) only patients with NSCLC, breast cancer, or colorectal cancer were included due to their standardized treatment procedures and the relatively small number of patients compared with other cancer types; (2) to assess targeted therapy-specific survival, patients receiving any other systematic treatments (e.g., chemotherapy, radiotherapy, endocrinotherapy) after sampling were excluded. PFS was defined as the duration from sampling time to first tumor progression.

All participants provided written informed consent before undergoing any study-related procedures. This study was performed in accordance with the Declaration of Helsinki.

**Sequencing probe construction**. Captured probes were first constructed for the entire regions of the most common driver genes across 12 solid tumor types to enhance detection sensitivity[47]. Next, genomic regions related to the effects of chemotherapy, targeted drugs, and immunotherapy per available clinical and preclinical research were added. Finally, frequently mutated regions recorded in the Catalogue of Somatic Mutations in Cancer (http://cancer.sanger.ac.uk/cosmic) and

TCGA (https://cancergenome.nih.gov/) were included. In total, the probes covered 1.09 Mbp of the genome. All included genes are shown in Supplementary Data 1.

**Targeted deep sequencing for circulating cfDNA and WBC DNA**. Peripheral blood samples were processed within 72 h after collection. The cellular constituent (mainly consisting of diverse lymphocytes) was separated via high-speed centrifugation ($2500 \times g$ for 10 min, followed by $16,000 \times g$ for 10 min). gDNA was extracted from the tumor tissue and WBCs using the QIAamp DNA Mini Kit and QIAamp DNA Blood Mini Kit (Qiagen, Hilden, Germany), respectively. cfDNA was extracted from plasma using the QIAamp Circulating Nucleic Acid Kit (Qiagen, Hilden, Germany). DNA concentration was estimated using a Qubit fluorometer and Qubit dsDNA High Sensitivity (HS) Assay Kit (Invitrogen, Carlsbad, CA, USA). The DNA fragment length was assessed using an Agilent 2100 Bioanalyzer and DNA HS Kit (Agilent Technologies, Santa Clara, CA, USA). Only cfDNA with >20 ng and ~170-bp fragments and gDNA with >20 ng and >500-bp fragments underwent subsequent processing.

gDNA were sheared into 200–250-bp fragments using a Covaris S2 instrument (Woburn, MA, USA), and indexed NGS libraries were prepared. For cfDNA, after end-repairing and A-tailing reactions, targeted adapters with unique identifiers were ligated to both ends of the double-stranded cfDNA fragments, followed by PCR to generate sufficient numbers of fragments prior to hybridization. Afterward, all libraries were hybridized to self-built probes, and DNA sequencing was performed using the MGISeq-2000 Sequencing System (BGI, Shenzhen, China) per the manufacturer's guidelines.

For data processing, raw reads were mapped to the reference human genome (hg19) using the Burrows–Wheel Aligner (http://bio-bwa.sourceforge.net/) with default parameters. Duplicate reads were identified and marked with Picard's Mark Duplicates tool (https://software.broadinstitute.org/gatk/documentation/tooldocs/4.0.3.0/picard_sam_markduplicates_MarkDuplicates.php) for gDNA data and clustered according to their unique identifier and the position of template fragments for cfDNA data. Local realignment and base quality recalibration were performed using The Gene Analysis Toolkit (https://www.broadinstitute.org/gatk/). Somatic Indels and SNVs were called using the MuTect2 algorithm (https://software.broadinstitute.org/gatk/documentation/tooldocs/3.8-0/org_broadinstitute_gatk_tools_walkers_cancer_m2_MuTect2.php).

Several filter procedures were executed after mutation calling. (1) Germline mutations with a ≥30% AF in both WBC gDNA and cfDNA were not analyzed and instead were filtered out. Exceptions were frameshifting Indels or truncating SNVs occurring in 1 of the 15 canonical CH genes. (2) Synonymous variants were filtered out. (3) Variants with low depth (i.e., <500× in cfDNA and tissue gDNA, 200× in WBC gDNA) were filtered out. Variants with <5 high-quality sequencing reads (mapqthres >30, baseqthres >30) for cfDNA/tissue gDNA and two high-quality sequencing reads for WBC gDNA were removed. (4) We built an in-house database of CH variants and cfDNA background noise from >10,000 cancer patients and healthy individuals. Variants were filtered if present in >1% samples in single-nucleotide polymorphism databases (dbsnp, https://www.ncbi.nlm.nih.gov/projects/SNP/; 1000G, https://www.1000genomes.org/; ESP6500, https://evs.gs.washington.edu/; ExAC, http://exac.broadinstitute.org/), or our in-house database. The remaining variants were identified as high-confidence somatic mutations. The 15 canonical genes associated with CH were *DNMT3A*, *TET2*, *ASXL1*, *PPM1D*, *TP53*, *JAK2*, *RUNX1*, *SF3B1*, *SRSF2*, *IDH1*, *IDH2*, *U2AF1*, *CBL*, *ATM*, and *CHEK2*[38].

Candidate fusion cfDNA fragments that were represented by the merging of overlapping paired-end reads were mapped to initial cfDNA fragments via unique barcoding and alignment information. Candidate fusion events with similar directionality and breakpoint proximity of soft-clipped reads were defined as the same cluster, and reference fragments were constructed. Only events with ≥2 unique post-realignment fragments were identified as high-confidence gene fusion. The Contra algorithm (http://contra-cnv.sourceforge.net) was used to detect copy number variants.

**Publicly available tissue sequencing database**. Tissue sequencing data were obtained from MSKCC[2] for comparison between ctDNA and tissue. A specific sequencing panel covering 468 genes was used in MSKCC. More than 60 cancer types and 10,000 patients with advanced cancer were included.

**Blood tumor mutation burden**. For the calculation of the bTMB, we applied three criteria for competent mutations: (1) somatic but not germline mutation; (2) located in coding region, nonsynonymous SNVs/Indels, including ±2 splice; and (3) a mutation allele frequency ≥0.5%. The bTMB was calculated as the number of competent mutations divided by the length of the panel-covered genomic region (1.09 Mb).

**MATH determination**. The MATH value of each allele was calculated from the median absolute deviation (MAD) and the median of its mutant AFs: MATH = $148.26 \times$ MAD/median[48]. The key purpose of the MATH value is to reflect the fluctuation range of AFs in the same sample.

**Estimation of co-occurring and mutually exclusive pathway alterations**. The co-occurrence and mutual exclusivity of one pathway with another pathway was estimated as the percentage of mutant samples/percentage of wild-type samples. Those with $q < 0.1$ (derived from the Benjamin–Hochberg procedure) are highlighted as significantly enriched co-pathways or mutually exclusive pathways.

**Estimation of mutation clonality**. To calculate mutation clonality, we first performed copy number transformation for the mutation of AFs according to a previous finding that, at high copy number, the relationship between the AF and copy number becomes log-linear for amplified mutations[43]: transformed AF = AF/$\log_2$(copy number). For mutations with a copy number loss, the transformation was ignored, and thus the transformed AF was equal to the original AF. Next, the normalized AF of each mutation was calculated as: transformed AF/maximal transformed AF in the same sample (range 0–1).

**Actionable mutation determination**. Evaluation of actionable targets was primarily based on the OncoKB database[49], which contains >3000 unique mutations, fusions, and copy number alterations in 418 cancer-associated genes. Briefly, the actionability of each mutation was appraised by the U.S. Food and Drug Administration (FDA) recognition and drug indications. There are four levels for sensitive markers. Level 1 was defined as a FDA-recognized biomarker for an FDA-approved drug in corresponding cancer types, Level 2 was defined as a standard-care biomarker for an FDA-approved drug, Level 3 was defined as a mutation with compelling clinical evidence supporting its function as a predictive biomarker of drug response, and Level 4 was defined as a mutation with compelling biological evidence supporting its function as a predictive biomarker of drug response. For Levels 2 and 3, the marker was labeled "A" if the cancer type matched that discussed in the corresponding evidence; otherwise, it was labeled "B". If one mutation was associated with multiple levels, it was determined as the highest level. Also, as one sample may carry a diversity of sensitive markers, the level of the sample was determined by the mutation with the highest level. In this study, Level 4 markers were ignored, as the clinical implication of these markers is unclear.

**Statistical analysis**. The consistency between two continuous variables was assessed using Pearson correlation analysis. The proportional compositions of two or more variables were compared using Chi-square or Fisher's exact tests. Mann–Whitney $U$ tests were used for the comparison of means between two datasets. Kaplan–Meier survival analysis was used to evaluate the association between bTMB/ctDNA AF and PFS. All statistical analyses were performed using SPSS 22.0 (IBM, Armonk, NY, USA). All tests were two sided, and $p$ values <0.05 were considered statistically significant.

**Reporting summary**. Further information on research design is available in the Nature Research Reporting Summary linked to this article.

## Data availability

Public databases used in the study include Catalogue of Somatic Mutations in Cancer (COSMIC, http://cancer.sanger.ac.uk/cosmic), The Cancer Genome Atlas (TCGA, https://cancergenome.nih.gov/), dbsnp (https://www.ncbi.nlm.nih.gov/projects/SNP/), 1000G (https://www.1000genomes.org/), ESP6500 (https://evs.gs.washington.edu/), ExAC (http://exac.broadinstitute.org/), and OncoKB (https://www.oncokb.org/). Sequencing data from MSKCC can be obtained from https://www.cbioportal.org/study/summary?id=msk_impact_2017. VCF files recording all raw mutational data (single-nucleotide variations, structural variations of DNA or chromosome, and copy number variations), the tumor classification of samples, demographic characteristics of patients, and the correspondence between samples and patients have been deposited in the European Variation Archive (https://www.ebi.ac.uk/ena/browser/view/PRJEB37947). The remaining data are available within the article, supplementary information or available from the authors upon request.

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

## Acknowledgements

We thank all our departmental colleagues for their constructive suggestions. This work was supported by Chinese Academy of Medical Sciences (CAMS) Innovation Fund for Medical Sciences (2018-I2M-1-002).

## Author contributions

Y.Z., Q.Z., and S.J. conceived and designed the experiment. Y.Y. and X.X. provided critical improvements to the conception of the work. Y.X. and Lifeng Li performed the experiments and analyzed the data. Y.X. wrote the manuscript. Y. Guan, K.Z., M.Z., and L.C. helped in data processing and interpretation. Y. Gong and Lin Li helped in improving the language and correcting grammar mistakes. All authors discussed the results and commented on the manuscript. All authors read and approved the final manuscript.

## Competing interests
The authors declare no competing interests.
