## [Peer Review File · Nature Communications]

Reviewers' Comments:

Reviewer #1:

Remarks to the Author:

Zhang et al. presented a comprehensive analysis of ctDNA in >10000 Chinese patients. This is probably the largest analysis of ctDNA so far, which covers a broad range of cancer types. Using this rich data, they studied clonal hematopoiesis, confirming their association with age. They also studied the mutational landscape of driver genes, and observed an overall correlation of mutational frequency with those in the MSKCC cohort. They assessed bTMB and pathway-based mutation frequency. Finally, they examined the therapeutic action ability revealed by ctDNA profiling. Overall, I think this is a highly valuable resource for the cancer research community, and it provides a benchmark dataset for assessing the utility of ctDNA and future applications. I applaud the authors for making such a large, diverse, informative dataset available.

I have the following comments to improve the study.

First, it should be very careful to explain the differences between the ctDNA cohort and the MSKCC cohort. Although the cohorts came from two different populations (Caucasian vs. Chinese), they are different in many aspects such as sample source, sequencing panel, detecting sensitivity, patient gender and age distributions. The authors seem to attribute to the observed difference to the origin of race, which may not be correct. The differences could be due to other confounding factors. I understand it may be hard to calibrate everything in such a comparison. But at least these factors should be discussed explicitly.

Second, the authors need to improve figure presentation. For example, Figure 1A, the heatmap color scheme is not informative at all (too blue); the size of each panel should be adjusted to make the heatmap larger and other panels smaller. Figure 2, the labels, "TMB stage" or "total cohort" are confusing; Figure 4, the color legend is not included. Just name a few. The data is great but need better visualization.

Third, the authors should make a serious effort to improve the language. There are many typos and grammar errors. For example, In Abstract, "Some mutant frequency bias, ..., implying the different driver pattern of Chinese and Caucasian patients." A professional editing service may be needed.

Fourth, to facilitate the community use of these data, I would suggest making a user-friendly data portal for this wonderful resource.

I would like to reveal my name: Han Liang

Reviewer #2:

Remarks to the Author:

The authors present an impressive survey of ctDNA NGS from >10k cancer patients. NGS was done with an in house panel w paired WBC NGS, though WBC was done at lower depth. Some clinical annotation was missing. The authors find expected mutation distributions, expected variation between shed and disease or stage.

I am impressed by the large cohort and the detailed presentation, BUT I am unsure what is novel here. The subclonality analysis is interesting but I unsure of a novel insight. My sense is the findings are largely confirmatory without a novel hypothesis tested.

One methods choice concerns me, which is the lower depth WBC sequencing. I thus presume there are low AF mutations likely from CHIP (<1% AF) which would be seen in the cfDNA but not screened

out in the CH analysis? This would lead to large numbers of low AF or subclonal mutations and could bias the overall cfDNA findings.

If the authors have clinical follow-up available, then one powerful analysis that is missing from the literature would be a study of how ctDNA shed is related to prognosis. I encourage the authors to consider novel added analyses to strengthen the impact.

Our responses to reviewer comments are provided below in italics with manuscript page numbers where relevant.

Reviewer #1: Expert in pan cancer genomics

Zhang et al. presented a comprehensive analysis of ctDNA in >10000 Chinese patients. This is probably the largest analysis of ctDNA so far, which covers a broad range of cancer types. Using this rich data, they studied clonal hematopoiesis, confirming their association with age. They also studied the mutational landscape of driver genes, and observed an overall correlation of mutational frequency with those in the MSKCC cohort. They assessed bTMB and pathway-based mutation frequency. Finally, they examined the therapeutic action ability revealed by ctDNA profiling. Overall, I think this is a highly valuable resource for the cancer research community, and it provides a benchmark dataset for assessing the utility of ctDNA and future applications. I applaud the authors for making such a large, diverse, informative dataset available.

I have the following comments to improve the study.

First, it should be very careful to explain the differences between the ctDNA cohort and the MSKCC cohort. Although the cohorts came from two different populations (Caucasian vs. Chinese), they are different in many aspects such as sample source, sequencing panel, detecting sensitivity, patient gender and age distributions. The authors seem to attribute to the observed difference to the origin of race, which may not be correct. The differences could be due to other confounding factors. I understand it may be hard to calibrate everything in such a comparison. But at least these factors should be discussed explicitly.

--Thank you for this constructive comment. We agree that race is only one possible factor and that other factors such as sample source, sequencing panel, detection sensitivity, and patient gender and age distributions may also explain discrepancies between ctDNA and MSKCC cohorts. In addition, we now cite a study reporting a relationship between EGFR prevalence and ethnicity/smoking history that supports this rationale. We added this content on Page 12

Lines 236-241 and Page 18 Lines 362-368.

Second, the authors need to improve figure presentation. For example, Figure 1A, the heatmap color scheme is not informative at all (too blue); the size of each panel should be adjusted to make the heatmap larger and other panels smaller. Figure 2, the labels, “TMB stage” or “total cohort” are confusing; Figure 4, the color legend is not included. Just name a few. The data is great but need better visualization.

--We apologize for our poor data visualization. Per your suggestion, we updated most figures and their corresponding legends to improve their presentation. Please see the attached files.

Third, the authors should make a serious effort to improve the language. There are many typos and grammar errors. For example, In Abstract,” Some mutant frequency bias, ..., implying the different driver pattern of Chinese and Caucasian patients.” A professional editing service may be needed.

--We apologize for the typos and grammatical errors in our manuscript. We reviewed the entire manuscript and used a professional editing service to improve its readability. The misleading sentence in the Abstract was corrected to “The landscape of putative driver genes revealed by ctDNA profiling was similar to that in a tissue-based database ($R^2=0.87$, $p<0.001$) but also showed some discrepancies, such as higher EGFR (44.8% versus 25.2%) and lower KRAS (6.8% versus 27.2%) frequencies in non-small cell lung cancer and higher TP53 frequency in hepatocellular carcinoma (53.1% versus 28.6%).”

Fourth, to facilitate the community use of these data, I would suggest making a user-friendly data portal for this wonderful resource.

--Thank you for this suggestion. We began constructing a related cloud platform around 1 year ago that contains tens of thousands of ctDNA results and an equal number of tissue results. Currently, the biggest challenge is the collection of general clinical characteristics and follow-up information. Its expected online launch will be toward the end of 2022. To facilitate community data use and ensure replicability, we deposited study-associated data in

the European Variation Archive (<https://www.ebi.ac.uk/eva/?eva-study=PRJEB37947>).

Reviewer #2: Expert in ctDNA genomics

The authors present an impressive survey of ctDNA NGS from >10k cancer patients. NGS was done with an in house panel w paired WBC NGS, though WBC was done at lower depth. Some clinical annotation was missing. The authors find expected mutation distributions, expected variation between shed and disease or stage.

I am impressed by the large cohort and the detailed presentation, BUT I am unsure what is novel here. The subclonality analysis is interesting but I unsure of a novel insight. My sense is the findings are largely confirmatory without a novel hypothesis tested.

--Thank you for your comments. Although ctDNA exhibits diverse application potentials, previous studies typically focus on single cancer types and have small sample sizes. For the first time, we performed profiling for real-world ctDNA detection from over 10,000 Chinese pan-cancer patients and made comprehensive comparisons with biopsy-based sequencing data. We believe our results are robust and practically meaningful. Recently, the FDA approved the first liquid biopsy next-generation assay for mutation profiling in advanced cancer patients with any solid malignant neoplasm, which will further enhance future applications of ctDNA.

There were two main purposes of this retrospective study. First, we sought to demonstrate that ctDNA is a practically applicable biomarker in genome-directed precision medicine by determining its detection sensitivity, concordance with tissue biopsy, and ability to identify actionable alterations. The conclusions drawn from this study are influenced by many factors and may be in better accord with the real-world scenario than well-designed clinical trials. Second, we sought to explore distinct uses of ctDNA in different cancer types. One novel finding is the extremely high ctDNA sensitivity and bTMB in both early and advanced SCLC. Although a few studies demonstrate the potential application of ctDNA in SCLC, including one conducted by our team^{1,2,3}, this study, for the first time, performed identical sequencing

and analytical processes to compare ctDNA profiles of SCLC with other cancer types. These findings are evidence of the potential utility of ctDNA in the early detection, genomic determination, and treatment management of SCLC. Another novel finding is the high sensitivity of c-kit actionable mutations in ctDNA from GIST patients, which could serve to improve treatment options and resistance monitoring.

In addition, this study is the first to demonstrate that ctDNA has a superior MATH value to tissue biopsy on the basis of a large-scale dataset of diverse cancer types, which quantitatively supports the common hypothesis that ctDNA performs better than single-region tissue sampling in identifying tumor heterogeneity. We then introduced a novel method to evaluate tumor clonal structure via ctDNA profiling. With the development of tumor evolutiology, clone-adjusted genomic markers exhibit promising application potential. For instance, targeting clonal trunk events is more effective than targeting subclonal events in reducing tumor burden⁴, and clonal TMB and neoantigen burden are closely associated with survival and immunotherapy efficacy^{5, 6}. On basis of the method used in this study, clonal structure can be determined despite tumor heterogeneity and tumor impurity introduced by single-region tissue sampling and thus could be used to develop optimized biomarkers associated with patient survival and therapeutic efficacy, such as clonal bTMB and clonal blood neoantigen burden. In addition, conventional pharmaceutical small-molecule approaches, such as EGFR tyrosine kinase inhibitors targeting single oncogenic drivers, can also benefit from ctDNA-directed clonal analysis. For instance, patients harboring clonal EGFR-sensitive mutations may exhibit better outcomes than those with subclonal mutations. We added this content to Page 18 Lines 371-386.

Most importantly, this study provides a rich resource for cancer studies and can serve as a benchmark to assess the utility of ctDNA and future applications. We hope the publication of this study enhances collaboration and data sharing among international cancer-related communities.

One methods choice concerns me, which is the lower depth WBC sequencing. I thus presume there are low AF mutations likely from CHIP (<1% AF) which would be seen in the cfDNA

but not screened out in the CH analysis? This would lead to large numbers of low AF or subclonal mutations and could bias the overall cfDNA findings.

--Thank you for this comment. Because the AFs of CH variants are significantly lower than that of non-CH mutations (Fig. 1), strict mutation filter strategies, such as the need for five high-quality sequencing reads (Page 24 Lines 492-494), dramatically decrease the detection likelihood of CH variants in cfDNA. We also built an in-house database containing CH variant and cfDNA background noise data from over 10,000 cancer patients and healthy individuals and filtered out cfDNA mutations present in >1% of samples in this database (Page 24 Lines 494-499). With prior knowledge that CH variants emerge equally in cancer patients and healthy individuals, we believe this step also filters out a considerable portion of CH variants.

We understand your concern about low-depth WBC sequencing. To provide stronger evidence, we performed additional research to examine the source of non-CH variants in cfDNA. We collected paired tumor tissue samples from 1,291 patients and conducted the same sequencing procedures. The median sequencing depth for tissue samples was 1,808× (range, 487- 5,525×). In plasma cfDNA from the same patients, 4,500 somatic mutations were detected in 976 samples (75.6%). Based on the detectability of each variant in different samples, we classified these variants into three categories: 316 WBC-matched variants (co-occurring in both plasma and WBCs, equal to CH variants), 4,023 biopsy-matched variants (co-occurring in both plasma and tissue biopsy), and 161 variants of unknown source (VUSOs; only occurring in plasma). On an individual level, 941 plasma samples (72.8%) harbored biopsy-matched variants, 214 (16.6%) harbored WBC-matched variants, and 124 (9.6%) harbored VUSOs (Fig. 1E). These results suggest that the vast majority of cfDNA somatic mutations filtered out by WBCs (96.2%, 4,032/4,193) could be verified in matched tumor tissue. Furthermore, the mutational fraction of 15 canonical CH-related genes was lower in VUSOs than in WBC-matched variants (27.5% versus 14.9%, Chi-square $p=0.002$; Fig. 1F), but AFs were similar between the two components and significantly lower than that of biopsy-matched variants (Fig. 1G). This indicates that although some VUSOs

may have been CH variants that were not screened out in WBC-matched analysis, a considerable fraction of VUSOs might be subclonal alterations missed in single-region sequencing or another (non-WBC) source of background noise (Page 7 Lines 124-145).

Fig. 1. Identifying CH variants in plasma cfDNA via matched WBC sequencing. (A) Percentage of plasma

samples with identified CH variants in different cancer types. The first row indicates the overall percentage of samples with any CH variants in different cancer types, and the other rows indicate the percentage of samples with CH variants in 15 canonical genes. Grey nubs indicate no CH variants were detected. (B) Density distribution of AFs of identified CH variants in cfDNA. Different panels indicate the distribution of different mutational types. (C) AFs of CH variants in cfDNA were significantly lower than non-CH mutations (Mann-Whitney U, *** $p < 0.001$). (D) The percentage of plasma samples with CH variants increases with the age of patients. (E) Relative fractions of biopsy-matched variants, WBC-matched variants, and VUSOs in each plasma cfDNA with matched tumor tissue and WBC sequencing. (F) Distribution of WBC-matched variants and VUSOs according to gene categories. Only canonical CH-related genes are shown. (G) Comparison of variant AFs among biopsy-matched variants, WBC-matched variants, and VUSOs (Mann-Whitney U test, *** $p < 0.001$). AF, allele frequency; CH, clonal hematopoiesis; VUSO, variant of unknown source; WBC, white blood cell; GIST, gastrointestinal stromal tumor; HCC, hepatocellular carcinoma; NSCLC, non-small cell lung cancer; SCLC, small cell lung cancer; UGI, upper gastrointestinal cancer.

Razavi et al.⁷ recently performed high-depth sequencing to reveal the sources of cfDNA mutations. Even with >60,000× raw depth, some cfDNA mutations, including 15 canonical CH genes, were missed in WBC-matched sequencing, in line with our findings (Page 18 Lines 357-360). Because economic cost was a major barrier for participants in this real-world study, we could not afford ultra-deep sequencing for all specimens. Based on the current depth range of cfDNA, we believe our WBC sequencing strategies are sufficient to yield robust information on tumor-derived mutations in cfDNA, and an expanded cfDNA sequencing database encompassing both healthy individuals and cancer patients further reduces interference from CH variants and non-CH noise. (Page 18 Lines 360-362).

If the authors have clinical follow-up available, then one powerful analysis that is missing from the literature would be a study of how ctDNA shed is related to prognosis. I encourage

the authors to consider novel added analyses to strengthen the impact.

--Thank you for this constructive suggestion. We retrospectively performed clinical follow-up for as many patients as possible. To ensure analytical standardization, we employed two additional exclusion criteria: (1) only NSCLC, breast cancer, and colorectal cancer patients were included in the survival analysis due to their standardized treatment procedures and the relatively small number of patients with other cancer types; (2) to assess targeted therapy-specific survival, patients receiving any other systematic treatment (e.g., chemotherapy, radiotherapy, endocrinotherapy) after sampling were excluded. PFS was defined as the duration from sampling time to the first tumor progression. Related content was added to the Materials and Methods section (Page 21 Lines 441-448).

Clinical follow-up was successfully completed for 137 patients receiving targeted therapies in accordance with ctDNA profiling, including 114 NSCLC, 17 breast cancer, and 6 colorectal cancer patients (Fig. 7A, Table S8). The vast majority of patients were stage IV, and PFS ranged from 4 to 1,198 days. Both mutation number and AF of plasma ctDNA exhibited moderately negative correlations with PFS (Fig. 7A, Fig. S11). The median PFS for patients with ≤ 2 mutations was 12.8 months, which was significantly longer than that for patients with > 2 mutations (median, 5.1 months; log-rank hazard ratio=2.648, 95% CI 1.838-3.815, $p < 0.001$; Fig. 7B). Likewise, patients with a ≤ 0.01 AF (median, 12.8 months) showed significantly longer PFS than those with a > 0.01 AF (median, 5.8 months; log-rank hazard ratio=2.190, 95% CI 1.561-3.073, $p < 0.001$; Fig. 7C). Related content was added to the Results section (Fig. 7, Table S7, Page 15 Lines 309-318).

Fig. 7. Circulating tumor burden determined by ctDNA profiling is associated with therapeutic prognosis.

(A) Overview of tumor types, clinical stages, PFS, ctDNA mutation numbers, and ctDNA AFs for 137 patients receiving targeted therapies guided by ctDNA profiling. (B) Patients with >2 mutations in ctDNA exhibited poorer PFS than those with ≤ 2 mutations in ctDNA. (C) Patients with >0.01 ctDNA AF exhibited poorer PFS than those with a ≤ 0.01 ctDNA AF.

Fig. S11. bTMB and ctDNA AF are negatively correlated with recurrence-free survival.

Because most patients in our cohort were in advanced stages and received first- or multi-line therapies before blood sampling, we only performed prognosis analysis with patients receiving systematic targeted treatments during or after sampling. Although the number of patients with follow-up data was much smaller than the total cohort, our results suggest that ctDNA level is closely related to the therapeutic prognosis of NSCLC, breast cancer, and colorectal cancer. Apart from the prognosis evaluation performed in this study, we published several papers revealing the association between postoperative ctDNA and disease-free survival for multiple cancer types, including gastric cancer⁸, hepatocellular carcinoma⁹, breast cancer¹⁰, and pancreatic cancer¹¹. Our unpublished data on operable colorectal cancer and ovarian cancer also suggest that ctDNA is a promising tool for predicting postoperative survival. In addition, numerous studies demonstrate that ctDNA profiling can predict patient prognosis in the context of diverse cancer types, different therapeutic regimens, and distinct disease stages^{12, 13, 14}. In summary, previous literature and our present findings support the prognostic utility of ctDNA.

The innate design of “real-world” sample collection restricts follow-up availability. To robustly validate the prognostic value of ctDNA, we recently established a prospective cohort

with standardized subject enrollment and clinical follow-up. Under the guidance of ctDNA identification, thousands of NSCLC patients will take targeted drugs, and routine follow-up will be performed every 3 months. Such efforts promise to strengthen the clinical impact of ctDNA detection.

References

1. Mohan S, *et al.* Profiling of Circulating Free DNA Using Targeted and Genome-wide Sequencing in Patients with SCLC. *J Thorac Oncol* **15**, 216-230 (2020).
2. Nong J, *et al.* Circulating tumor DNA analysis depicts subclonal architecture and genomic evolution of small cell lung cancer. *Nat Commun* **9**, 3114 (2018).
3. Devarakonda S, *et al.* Circulating Tumor DNA Profiling in Small-Cell Lung Cancer Identifies Potentially Targetable Alterations. *Clin Cancer Res* **25**, 6119-6126 (2019).
4. Lee JK, *et al.* Spatiotemporal genomic architecture informs precision oncology in glioblastoma. *Nat Genet* **49**, 594-599 (2017).
5. McGranahan N, Swanton C. Neoantigen quality, not quantity. *Sci Transl Med* **11**, (2019).
6. McGranahan N, *et al.* Clonal neoantigens elicit T cell immunoreactivity and sensitivity to immune checkpoint blockade. *Science* **351**, 1463-1469 (2016).
7. Razavi P, *et al.* High-intensity sequencing reveals the sources of plasma circulating cell-free DNA variants. *Nat Med* **25**, 1928-1937 (2019).
8. Yang J, *et al.* Deep sequencing of circulating tumor DNA detects molecular residual disease and predicts recurrence in gastric cancer. *Cell Death Dis* **11**, 346 (2020).
9. An Y, *et al.* The diagnostic and prognostic usage of circulating tumor DNA in operable hepatocellular carcinoma. *Am J Transl Res* **11**, 6462-6474 (2019).
10. Li S, *et al.* Circulating Tumor DNA Predicts the Response and Prognosis in Patients With Early Breast Cancer Receiving Neoadjuvant Chemotherapy. 244-257 (2020).
11. Jiang J, *et al.* Circulating Tumor DNA as a Potential Marker to Detect Minimal Residual Disease and Predict Recurrence in Pancreatic Cancer. **10**, (2020).
12. Tie J, *et al.* Circulating tumor DNA analysis detects minimal residual disease and predicts recurrence in patients with stage II colon cancer. *Sci Transl Med* **8**, 346ra392 (2016).
13. Seremet T, *et al.* Undetectable circulating tumor DNA (ctDNA) levels correlate with favorable outcome in metastatic melanoma patients treated with anti-PD1 therapy. *J Transl Med* **17**, 303 (2019).
14. Lee JH, *et al.* Pre-operative ctDNA predicts survival in high-risk stage III cutaneous melanoma patients. *Ann Oncol* **30**, 815-822 (2019).

Reviewers' Comments:

Reviewer #2:

Remarks to the Author:

I very much appreciate the revisions and am impressed by the responsiveness of the investigators and the new data included.

For clarity, I encourage them to revisit this sentence:

"Furthermore, the mutational fraction of 15

147 canonical CH-related genes was lower in VUSOs than in the WBC-matched variants (27.5%
148 versus 14.9%, Chi-square $p=0.002$; Fig. 1F)."

I believe they are saying that of ### WBC-matched variants, 27% were in the 15 canonical CH-related genes, while of the ### VUSOs a smaller proportion (14%, $p=##$) were in these CH-related genes? Figure 1F doesn't actually show this analysis. Maybe Figure 1F can be clarified or better articulated - I don't see the 15 CH-related genes listed in Figure 1F. I believe I understand the intended analysis.

Our responses to reviewer comments are provided below in italics with manuscript page numbers where relevant.

Reviewer #2 (Remarks to the Author):

I very much appreciate the revisions and am impressed by the responsiveness of the investigators and the new data included.

For clarity, I encourage them to revisit this sentence:

"Furthermore, the mutational fraction of 15 canonical CH-related genes was lower in VUSOs than in the WBC-matched variants (27.5% versus 14.9%, Chi-square $p = 0.002$; Fig. 1F)."

I believe they are saying that of ### WBC-matched variants, 27% were in the 15 canonical CH-related genes, while of the ### VUSOs a smaller proportion (14%, $p=##$) were in these CH-related genes? Figure 1F doesn't actually show this analysis. Maybe Figure 1F can be clarified or better articulated - I don't see the 15 CH-related genes listed in Figure 1F. I believe I understand the intended analysis.

--Thank you for your kind remarks. We admit that the sentence and corresponding Figure may lead to ambiguity and acknowledge that CH variants are still present even after WBC-matched screening. We have provided the total list of 15 CH-related genes in Page 24 Line 503-504, and only 10 genes are present in WBC-matched variants and VUSOs in the subgroup analysis. In order to clarify our analysis process, we have changed the sentence to the following form:

"Totally 10 (DNMT3A, TP53, TET2, ASXL1, PPM1D, ATM, JAK2, SF3B1, CHEK2, CBL) out of 15 canonical CH-related genes (see Materials and Methods) were involved in either WBC-matched variants or VUSOs. Of 316 WBC-matched variants, 27.5% were in the 10 canonical CH-related genes, while of the 161 VUSOs a smaller proportion (14.9%, Chi-square $p = 0.002$) were in these CH-related genes (Fig. 1F)."

Meanwhile, we have changed the next sentences to the following form:

“The AFs were similar between the two aforementioned components and significantly lower than that of the biopsy-matched variants (Fig. 1G). It seemed that CH variants might still be present even after WBC-matched screening. However, the proportion would be limited compared with tumor-derived variants. Subclonal alterations missed in single-region sequencing or other (non-WBC) source of background noise might constitute the remaining proportion of VUSOs.”

These modifications can be found in Page 8 Line 143-153.